# SAXS Investigation of the Effect of Freeze/Thaw Cycles on the Nanostructure of Nafion^®^ Membranes

**DOI:** 10.3390/polym14204395

**Published:** 2022-10-18

**Authors:** Ruslan M. Mensharapov, Nataliya A. Ivanova, Dmitry D. Spasov, Sergey A. Grigoriev, Vladimir N. Fateev

**Affiliations:** 1National Research Center “Kurchatov Institute”, 1, Akademika Kurchatova sq., 123182 Moscow, Russia; 2National Research University “Moscow Power Engineering Institute”, 14, Krasnokazarmennaya st., 111250 Moscow, Russia; 3HySA Infrastructure Center of Competence, Faculty of Engineering, North-West University, Potchefstroom 2531, South Africa

**Keywords:** Nafion membrane, SAXS, freeze/thaw cycling, surface layer, SiO_2_ nanoparticles

## Abstract

In this study, we performed small-angle X-ray scattering (SAXS) to investigate the structure of Nafion^®^ membranes. The effect of freeze/thaw (F/T) cycles (from ambient temperature down to −40 °C) on the membrane nanostructure was considered for the first time. The SAXS measurements were taken for different samples: a commercial Nafion^®^ 212 membrane swollen in water and methanol solution, and a water-swollen silica-modified membrane. The membrane structure parameters were obtained from the measured SAXS profiles using a model-dependent approach. It is shown that the average radius of water channels (*R_wc_*) decreases during F/T cycles due to changes in the membrane structure as a result of ice formation in the pore volume after freezing. The use of water-methanol solution (methanol content of 20 vol.%) for the membrane soaking prevents changes in the membrane structure during F/T cycles compared to the water-swollen membrane. Modification of the membrane surface with silica (SiO_2_ content of 20 wt.%) led to a redistribution of water in the membrane volume and resulted in a decrease in *R_wc_*. However, *R_wc_* for the modified membrane did not decrease with the increasing number of F/T cycles due to the involvement of SiO_2_ in the sorption of membrane water and, therefore, the prevention of ice formation.

## 1. Introduction

Hydrogen technologies are rapidly developing in the world. Proton exchange membrane fuel cells (PEMFCs) are one of the most promising hydrogen electrochemical systems. PEMFCs have a number of important advantages: fast start-up, small size per unit of generated power, and low operating temperatures. These features make PEMFCs suitable for portable devices, vehicles, backup power supply systems, and electricity generation in geographically remote regions. However, to ensure efficient operation and the cost effectiveness of such devices, it is important to increase the stability of PEMFCs during operation under changing external conditions, in particular, cyclic temperature changes.

During PEMFC operation under variable temperatures, thermal expansion/contraction of membrane-electrode assembly (MEA) components takes place. This effect tends to be strongest when PEMFC operates under subzero temperature conditions, which may be accompanied by the formation of ice from excess water contained in the MEA volume [1,2,3]. The cyclic formation and melting of ice occurs during repeated shutdowns and start-ups of PEMFCs at subzero ambient temperatures. These operating conditions may lead to negative effects, such as mechanical damage to the PEMFC electrodes and membrane [4,5,6,7].

The most widely used PEM is Nafion^®^ (DuPont), and its molecular structure is:(1)$$\begin{array}{c} -{\left[\mathrm{CF}-{\mathrm{CF}}_{2}-{\left({\mathrm{CF}}_{2}-{\mathrm{CF}}_{2}\right)}_{n}\right]}_{m}-\\ \;\;\;\;\;\;|\\ \;\;\;\;\;O-\mathrm{CF}-{\mathrm{CF}}_{2}-O-{\mathrm{CF}}_{2}-{\mathrm{CF}}_{2}-{\mathrm{SO}}_{3}^{-}H^{+}\\ \;\;\;\;\;\;\;\;\;\;\;\;\;\;\;|\\ {\;\;\;\;\;\;\;\;\;\;\;\;\;\;\mathrm{CF}}_{3} \end{array}$$

Nafion^®^ membrane is composed of hydrophobic polytetrafluoroethylene backbone and side chains with hydrophilic sulfonic acid groups, which form ionic domains in wet membrane. Due to the higher mobility of short side chains compared to the polymer backbone, in addition to the formation of large defects during temperature cycling, reorganization of the PEM nanostructure is also possible [8]. Such structural changes may lead to the formation of isolated ionic domains that do not further participate in the formation of water channels and proton transport. Additionally, mechanical deformation and additional stresses when the membrane is frozen may accelerate degradation of the membrane [9].

To mitigate the negative impact of subzero temperatures on the PEMFC performance efficiency, a number of approaches are used for the prevention of ice formation. The most popular approach used for removing excess moisture from MEA is dry gas purging [10,11,12,13,14]. This method can completely prevent water crystallization. However, it requires additional equipment, complicates the shutdown/start-up strategy, and significantly increases the time to start up PEMFC and achieve operational characteristics. More promising methods are the use of alcohol mixtures instead of water for the membrane swelling [15,16,17,18] or the use of hydrophilic additives involved in the binding of free water [19,20,21]. These approaches make it possible to significantly increase the service life of the MEA components under conditions of subzero temperatures, and to start up and shutdown PEMFCs without additional stages and equipment.

When a methanol mixture is added for membrane swelling, additional processes may occur, affecting the PEMFC performance efficiency. As has been reported before [22,23], methanol molecules can integrate into the polymer backbone, leading to deterioration of the membrane strength and methanol crossover. A methanol content in solution of 10–20 vol.% was proposed as the optimal one [17]. At lower methanol concentrations, the MEA stability with regard to freezing is insufficient while at higher concentrations, the Nafion^®^ membrane dissolves, its strength decreases, and the ionic resistivity tends to increase.

Inorganic hydrophilic compounds are widely used as water sorbents in the development of PEMFCs that are stable at low humidity; both catalytic layers [24,25,26,27,28] and membranes [21,29,30,31,32,33] can be modified. Membrane modification results in a greater water retention effect, but introduction of a sorbent leads to an increase in the ionic resistivity, resulting in PEMFC performance degradation [34]. To reduce the influence of the modifier, the membrane can be partially modified, in particular by the incorporation of sorbent particles into the surface layer. According to our previous research [33], the addition of a modifier on the cathode side of the membrane provides a better water retention ability. Nevertheless, the influence of the modifier on the membrane structure is still not fully understood. Thus, taking into account the PEMFC operating strategies under subzero temperatures, it is important to evaluate the effect of introducing alcohol or a modifier on the membrane structure in order to optimize the final properties of the PEMFC as a whole.

Small-angle X-ray scattering (SAXS) is one of the most efficient techniques for conducting accurate studies of the nanostructure of phase-separated polymer materials, such as Nafion^®^ membrane [35,36,37,38]. The SAXS profile *I(q)* of the water-swollen Nafion^®^ membrane has two scattering peaks, reflecting two main features of the membrane structure [39,40,41]. The scattering peak located in the low-*q* region (*q* < 0.7 nm^−1^) is associated with lamellar crystalline domains formed by perfluorinated hydrophobic chains. The lamellar structure is formed during membrane manufacturing and is crucial for the final mechanical properties of PEM. The second scattering peak located in the high-q region (1 < *q* < 2 nm^−1^) is attributed to the ionic cluster domains that appear when the membrane is swollen. The outer shell of the ionic domain consists of sulfonic groups and short side chains of the polymer, and the core contains water. When there is enough moisture, the ionic domains form a network of water channels that connect opposite sides of the membrane and provide proton transport [42]. Thus, ionic domains determine the proton conductivity of the membrane, and the evaluation of their geometric characteristics is an important task.

The geometric parameters of the ionic domains can be estimated from the measured SAXS profiles through calculation of the invariants using the model-independent Porod analysis. According to [37], the value of the ionic domain radius for the water-swollen Nafion^®^ 117 membrane was 2.4 nm. Various models can also be used to determine the values of structural parameters from the SAXS profile analysis. In particular, the core-shell model was used for SAXS profile analysis in [38,43,44]. According to this model, ionic domains can be represented as a spherical central core consisting of water and an outer shell containing several sulfonic groups. A more accurate core-shell model, which agrees with the Nafion^®^ membrane properties, was proposed in [38,45]. According to this model, ionic cluster domains represent long parallel cylindrical channels filled with water.

The objective of our work was to study the influence of F/T cycles on the Nafion^®^ nanostructure using the SAXS technique. The membrane nanostructure was obtained by optimizing the model parameters to achieve the best match between the calculated and measured SAXS profiles. The model of parallel cylindrical water channels was used in the calculations. The SAXS profiles of water-swollen Nafion^®^ 212 membranes before and after 15 and 30 F/T cycles were measured. A correlation between the structural parameters and the number of F/T cycles was established. The stability of the membrane swollen in the water-methanol mixture with regard to subzero temperatures was studied using the SAXS method. The SAXS profiles for samples of water-swollen and silica-modified membrane before and after 30 F/T cycles were measured. The influence of the addition of the modifier on the structural properties of the membrane and its stability under subzero temperatures was evaluated.

## 2. Materials and Methods

### 2.1. Preparation of Nafion^®^ Membranes

A commercial 50-μm-thick Nafion^®^ 212 membrane (DuPont, Wilmington, DE, USA) was chosen for this study. All membrane samples were converted to the H^+^ form by soaking in a 10 wt.% aqueous solution of HNO_3_ (Uralchem, Moscow, Russia) at 90 °C for 1 h, followed by double washing with deionized (DI) water under the same conditions [17].

A series of F/T cycles in the temperature range from −40 to 20 °C were used to investigate the effect of thermal cycling on the nanostructure of membrane samples fully swollen in water. The F/T cycling procedure is described in detail in [17]. To evaluate the effect of methanol addition, the membrane was soaked in a water-methanol (Metafrax, Gubakha, Russia) mixture (20 vol.% of methanol) before the F/T cycling.

Samples of silica-modified membranes were prepared by air-spraying silica/Nafion^®^ suspension onto the membrane surface from one side. The suspension was prepared by mixing the highly dispersed hydrophilic silica particles (ORISIL 300, Orisil-Kalush, Ukraine) with a size of 2–40 nm and Nafion^®^ ionomer in isopropanol/water solution followed by ultrasonication at room temperature for 30 min. The average thickness of the surface layer was about 15 μm and the content of the modifier in the membrane was 10 wt.% (the content in the surface layer was 40 wt.%). The obtained silica-modified samples were soaked in DI water before the F/T cycling and SAXS measurements.

### 2.2. Small-Angle X-ray Scattering

The SAXS experiments were performed at the BioMUR beamline at the Kurchatov Synchrotron Radiation Source (Moscow, Russian Federation) [46]. The beam energy was 8 keV (*λ* = 1.445 Å) and the beam size was 500 × 350 μm. As a detector, the 2D Pilatus3 1M system (DECTRIS, Switzerland) was used. The SAXS experiments were performed for the scattering vector range *q* = 0.02–6 nm^−1^, where *q* = 4π sin(*θ*)/*λ*, and *θ* is the scattering angle. The SAXS profiles were measured in constant temperature and moisture conditions.

### 2.3. Analysis Methods

The SAXS profiles were simulated and fitted to experimental ones using the SASfit code [47]. The inverse core-shell cylinders shown in Figure 1 were used as model objects to simulate the electron density of the membrane nanostructure. The core with a radius *R_wc_* corresponded to the water channel, and the shell with a thickness *h* was associated with the sulfonic groups and short side chains of the polymer. Depending on the membrane sample, the core consisted of water or a water-methanol mixture. Cylindrical ionic domains were located parallel to each other in the amorphous polymer matrix.

According to the model used, the SAXS intensity can be expressed as:(2)$$\begin{array}{c} I\left(q\right)\;\propto \;{\left(K\left(q,\;\eta_{wc}\;–\;\eta_{s},\;R_{wc},\;L\right)+K\left(q,\;\eta_{s}\;–\;\eta_{m},\;R_{wc}+h,\;L\right)\right)}^{2},\\ K\left(q,\;∆\eta ,\;R,\;L\right)=2\pi R^{2}L\Delta \eta \frac{J_{1}\left(qR\right)}{qR}, \end{array}$$
where *η_wc_*, *η_s_*, and *η_m_* is the scattering length density (SLD) of the water channel, side sulfonic chains, and amorphous polymer matrix, respectively; *L* is the length of the cylinder; *h* is the thickness of the cylindrical shell; ∆*η* is the difference between two SLDs; and *J*_1_ is the regular cylindrical Bessel function of first order [47].

To achieve better agreement between the experimental and model SAXS data, a structure factor was introduced. Seven parallel ion domains form a hexagonal cluster (see Figure 1b) with a fixed distance *a* between the centers of adjacent cylindrical channels [38]. Additionally, a normal distribution with a standard deviation *σ_R_* was used for the radius of the cylindrical channels.

The initial parameters of the model are presented in Table 1. The final values of the water channels radius, shell thickness, and distance between ionic domains of the hexagonal cluster were obtained by optimizing the parameters to achieve the best agreement between the SASfit-modeled and measured SAXS profiles.

To compare the model and silica-modified membrane data, the contribution of amorphous SiO_2_ to the scattering intensity was preliminarily subtracted from the SAXS profiles.

## 3. Results and Discussion

Figure 2 shows the SAXS profile of the fully water-swollen Nafion^®^ 212 membrane. The SAXS profile has two peaks. The first peak (*q* = 0.59 nm^−1^) is associated with the lamellar polymer crystallites, and the second one (*q* = 1.82 nm^−1^) is related to water channels.

The cylindrical water channel model was used to obtain the SAXS profile (see Figure 3). The calculated data are in good agreement with the experimental profile in the ionic domain peak region.

The model fitting results and structural parameters obtained in other studies are compared in Table 2. The presented parameters have close values; the discrepancies can be explained by the differences in the models used, types of PEMs, and their water uptake capacity.

To investigate the effect of subzero temperatures on the initial water-swollen Nafion^®^ 212 membrane, membranes after 15 and 30 F/T cycles were prepared. The experimental and modeled SAXS profiles of these samples are shown in Figure 4. For all membranes, the peaks of the lamellar structure (see Figure 4a) have the same position and are observed in the region *q* = 0.59 nm^−1^, which corresponds to the correlation distance between matrix crystallites of 10.6 nm, and is in good agreement with the results obtained in other studies. The lack of changes in the lamellar structure indicates a small effect of F/T cycles on the membrane mechanical properties.

The SAXS profiles in the region of the water channel peak are shown in Figure 4b, and the radii of water channels obtained from the model are given in Table 3. It was found that the water channel radius decreases with the increasing cycle number. This effect can be explained by a change in the channel structure as a result of the rearrangement of short chains and formation of isolated ionic domains blocked from interacting with water. The blocking of individual sulfonic groups is similar to their loss and an increase in the equivalent mass of the ionomer, which reduces the size of the water channels [43]. These structural changes in the membrane, along with the degradation and delamination of the catalytic layer [1,2,3,4,5,6,7], make an additional contribution to the increase in ionic resistance and deterioration of the PEMFC performance efficiency during F/T cycles.

Figure 5 shows the SAXS profile for the membrane swollen in a water-methanol mixture and exposed to 30 F/T cycles. The values of the water channel sizes are presented in Table 4. Due to the lower crystallization temperature, methanol acts as antifreeze [17,18] and prevents the formation of ice from the excess moisture. The solidification temperature of a water-methanol solution (20 vol.% of methanol) is about −13 °C [51], but the small size of ionic domains and the presence of sulfonic groups reduces the freezing point of a water-methanol mixture in the volume of the membrane channels to lower values. The coincidence of the SAXS profiles for the two membranes indicates the lack of structural changes after F/T cycles in the presence of methanol. This result points to the protective properties of methanol and the main contribution of ice formation inside the water channels to structural changes in the membrane at subzero temperatures. Although methanol can dissolve the membrane and become incorporated into the hydrophobic matrix [22], at a low methanol content of 20 vol.%, this effect is insignificant as there are no changes in the lamellar structure peak of the SAXS profile. Slight differences in the peak intensity can be explained by the difference in the solvents used and the irregularity of the thickness of the samples under study.

Figure 6 shows SAXS profiles of the silica-modified membranes. The water channel radii obtained from the model fitting are shown in Table 5. The obtained data indicate a decrease in the average size of the water channels for the silica-modified membrane compared to the initial Nafion^®^ 212. This effect can be explained by the redistribution of water in the membrane volume and its flow to the surface layer [33], resulting in a decrease in the radius of ionic domains in the membrane volume. The contribution of the modified layer to the scattering intensity in the high-q region was low due to the high silica content (40 wt.%) and relatively small thickness of the surface layer (about 15 µm). After 30 F/T cycles, an increase in the size of the channels for the modified membrane was observed. The increase in the channel radius may be associated with the backflow of water from the modified surface to the membrane volume as a result of partial aggregation and dissolution of the modifier particles during the F/T cycles [52]. Nevertheless, the lack of a decrease in the water channel size after a series of F/T cycles indicates that the surface modification of the membrane with hydrophilic SiO_2_ particles has a positive effect on its stability under subzero temperature cycling.

The high contribution of SiO_2_ nanoparticles to the scattering intensity in the low-q region did not indicate the position of the scattering peak associated with the lamellar polymer crystallites. Thus, the effect of F/T cycling on the lamellar structure of the silica-modified membrane requires further research.

Additionally, it should be pointed out that during freezing, free water may flow out from the channels to the membrane surface [20], and the addition of the modifier into the surface layer can prevent ice formation.

## 4. Conclusions

The effect of F/T cycles on the Nafion^®^ 212 membrane nanostructure was characterized through SAXS measurements. Using an inverse cylindrical core-shell model, the SAXS profiles in the *q* region of the ionic domains peak were calculated, and the sizes of the ionic domains were obtained by fitting the model parameters to the experimental data.

The F/T cycling did not affect the lamellar structure of the membrane but led to a decrease in the size of the water channels, which can be explained by a rearrangement of short-side chains. Such structural changes in the membrane may provide an additional contribution to the increase in ionic resistivity and reduction in the PEMFC performance efficiency.

The SAXS profiles obtained for the membrane soaked in the 20 vol.% water-methanol mixture after 30 F/T cycles and for the initial water-soaked membrane coincided over the entire *q* measurement range. Thus, ionic domain rearrangement and the incorporation of alcohol molecules into the structure of the hydrophobic matrix were not observed.

The introduction of SiO_2_ particles into the surface layer of the membrane led to a decrease in the size of the channels as a result of the redistribution of water in the membrane volume. The F/T cycling resulted in an increase in the size of the channels for the silica-modified membrane, which can be explained by further redistribution of water from the surface to the membrane volume due to the modifier loss. However, a positive effect of the addition of hydrophilic SiO_2_ particles on the membrane stability under subzero temperatures was indicated.

## Figures and Tables

**Figure 1 polymers-14-04395-f001:**
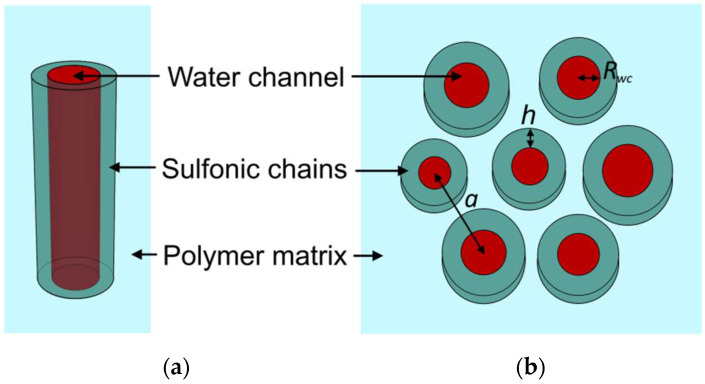
Model schematic of a separate cylindrical ionic domain (**a**) and hexagonal cluster (**b**).

**Figure 2 polymers-14-04395-f002:**
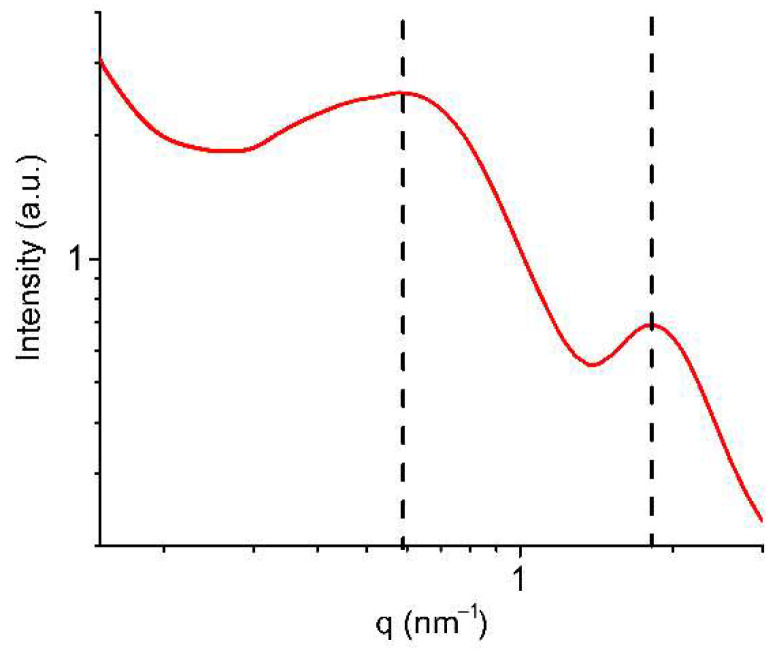
SAXS profile of the water-swollen Nafion^®^ 212 membrane.

**Figure 3 polymers-14-04395-f003:**
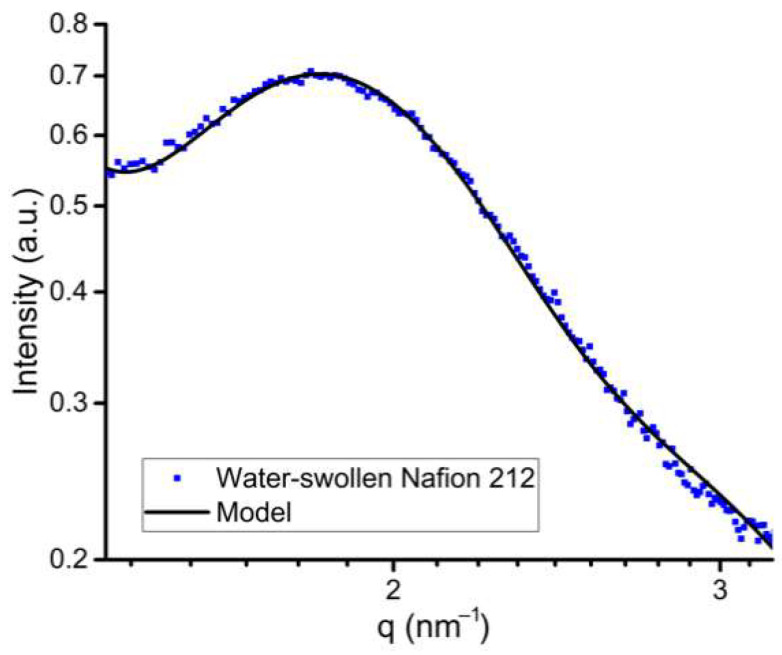
Experimental and model SAXS profiles in the ionic domain peak region.

**Figure 4 polymers-14-04395-f004:**
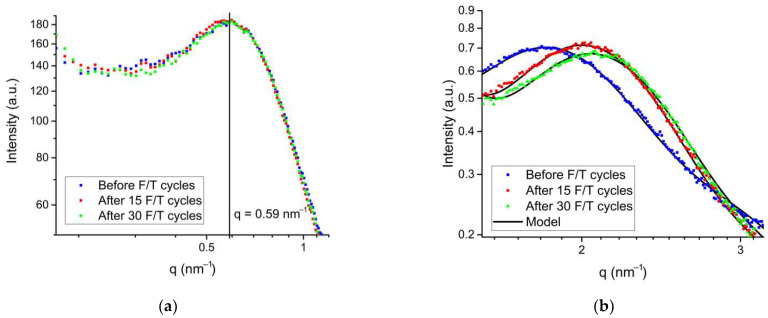
Experimental and model SAXS profiles of the membranes before and after 15 and 30 F/T cycles in the region of the lamellar structure peak (**a**) and ionic domain peak (**b**).

**Figure 5 polymers-14-04395-f005:**
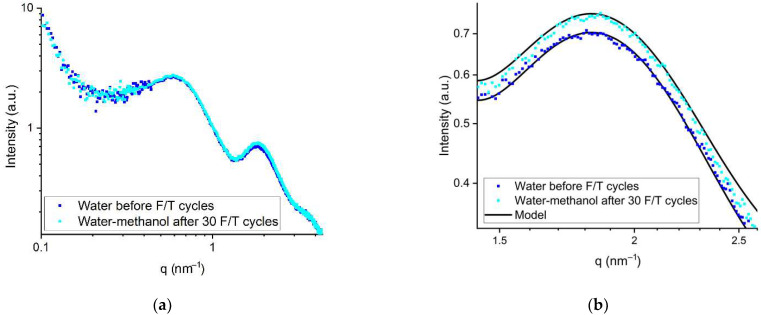
SAXS profiles of the membranes swollen in water and in a water-methanol mixture before and after 30 F/T cycles, respectively (the entire measuring range (**a**) and the region of the ionic domains peak (**b**)).

**Figure 6 polymers-14-04395-f006:**
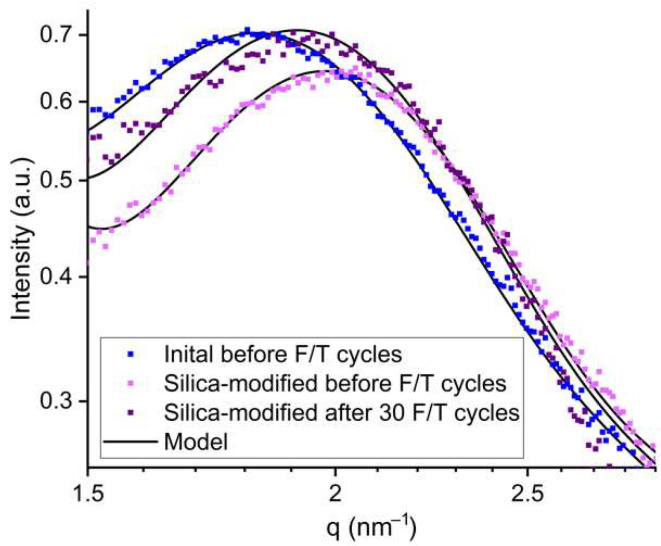
Experimental and model SAXS profiles of the initial water-swollen and silica-modified membranes.

**Table 1 polymers-14-04395-t001:** Model parameters of the swollen Nafion^®^ membrane [37,38,48].

Model Parameters	Value
*R_wc_*	<3 nm
*h*	<1 nm
*a*	~4 nm
SLD of water	9.412 × 10^10^ cm^−2^
SLD of sulfonic groups and side chains	16.147 × 10^10^ cm^−2^
SLD of perfluorinated polymer matrix	15.443 × 10^10^ cm^−2^
SLD of methanol	9.566 × 10^10^ cm^−2^
SLD of SiO_2_	15.870 × 10^10^ cm^−2^

**Table 2 polymers-14-04395-t002:** Structural parameters of the ionic domains of the Nafion^®^ membrane swollen in water: the radius of the ionic domains (*R_wc_*), standard deviation of the radii distribution (*σ_R_*), thickness of the domain shell (*h*), and the distance between ionic domains (*a*).

Membrane	*R_wc_*, nm	*σ_R_*	*h*, nm	*a*, nm
Nafion^®^ 212	2.64	0.48	0.7	3.8
Nafion^®^ 117 [37]	2.44	0.25	–	4.7
Nafion^®^ 212 [45]	2.42	–	0.7–1.2	3.8
Nafion^®^ 212 [49]	2.30	–	0.3	–
Nafion^®^ 212 [50]	2.10	–	–	–

**Table 3 polymers-14-04395-t003:** The size of water-swollen Nafion^®^ 212 ionic domains before and after F/T cycles.

Membrane	*R_wc_*, nm	*σ_R_*
Before F/T cycles	2.64	0.48
After 15 F/T cycles	2.41	0.47
After 30 F/T cycles	2.36	0.47

**Table 4 polymers-14-04395-t004:** The size of ionic domains for the Nafion^®^ 212 membrane swollen in water and a water-methanol mixture.

Membrane	*R_wc_*, nm	*σ_R_*
Water before F/T cycles	2.64	0.48
Water-methanol after 30 F/T cycles	2.65	0.47

**Table 5 polymers-14-04395-t005:** Size of the ionic domains of the water-swollen Nafion^®^ 212 and silica-modified membrane before and after F/T cycles.

Membrane	*R_wc_*, nm	*σ_R_*
Initial before F/T cycles	2.64	0.48
Silica-modified before F/T cycles	2.44	0.47
Silica-modified after 30 F/T cycles	2.55	0.47

## Data Availability

Not applicable.

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
