# Peer review of "SAXS Investigation of the Effect of Freeze/Thaw Cycles on the Nanostructure of Nafion® Membranes"

_polymers, 2022, doi:10.3390/polym14204395_

Round 1

Reviewer 1 Report

The paper reports about  interesting results which can be useful to scientists working in the field of membranes for electrolytic application.vdetrailed structural information not easy to reach  and a description of the   membrane multiphase  organization thanks to the application of SAXS to analyze this material. 

This is certainly the strong aspect of the paper which deserves to be considered for publication.

In my opinion  as a scientific paper in an international journal the paper could be improved  in two direction:

  # The description of the materials examined  should be provided dawn to molecular level  and the submiicroscopic  features expalined in details  with connection to the polymer structure and water interaction.

# The objective of the paper is claimed as "to study the influence of F/T cycles on the Nafion® 110 nanostructure by the SAXS technique". This is what the author  wish to do and not what they want to demonstrate as a contribution to the scientific community. How  do their results contribute to the understanding and better formulation of the membranes for a more efficient elctrolytic process ?

Reviewer 2 Report

The authors presented an important study on the structure change of Nafion membrane and Nafion/SiO2 membrane under a cycling condition that is relevant to the PEMFCs application. Overall, this is a solid study with good experimental designs and interesting results. However, there are some questions that need to be addressed before the manuscript can be accepted for publication:

1. Authors should add the meanings of the terms used in equation (1) and those listed in Table 1, 2.

2. Line 209, authors mentioned the degradation and the delamination of catalyst layer after the cycles, has this been observed or previously reported ? Please present the relevant data or citations to support this statement.

3. Authors are suggested to discuss more on the reason why the the mixed water/methanol solution helps to maintain the ionic domain structure after the F/T cycles.

4. Why does the SiO2 particles coating on the surface of Nafion membrane decrease its ionic domain size? It seems to be an effect of the membrane preparation process, otherwise I would expect the ionic domain size to increase with SiO2 particles added as the hydrophilic silica may facilitate the formation of larger water channels/ionic domains.

5. How does the crystalline domain size change after incorporating SiO2 particles in the Nafion membrane? Does the Nafion/SiO2 composite membrane have a lower crystallinity or worse crystallinity after the C/T cycles? The results should have been added.

6. What is the size of SiO2 particles and how does this affect the ionic domain size? 

Reviewer 3 Report

The manuscript entitled “SAXS investigation of freeze/thaw cycles effect onto the  nanostructure of Nafion ® membranes”. Some issues to be addressed will improve the quality of the manuscript. Therefore, I recommend this work could be published after the major revision

1.      Is it appropriate for the author to include the novelty of this review article in the abstract?

2.       The English composition may use some work. To reduce grammatical errors, the authors should proofread the content thoroughly.

3.       All references cited in the article must be cited in the text, and vice versa.

4.       This study issue has been extensively researched, and numerous studies have been conducted. Please include a comparison table for the reader's convenience of understanding.

5.      Please keep the conclusion brief and to the point.
